# Enhancing Night and Day Circadian Contrast through Sleep Education in Prediabetes and Type 2 Diabetes Mellitus: A Randomized Controlled Trial

**DOI:** 10.3390/biology11060893

**Published:** 2022-06-10

**Authors:** Cristina García-Serrano, Jesús Pujol Salud, Lidia Aran-Solé, Joaquim Sol, Sònia Ortiz-Congost, Eva Artigues-Barberà, Marta Ortega-Bravo

**Affiliations:** 1Balaguer Primary Care Centre, Institut Català de la Salut (ICS), 25600 Lleida, Spain; jpujol.lleida.ics@gencat.cat (J.P.S.); laran.lleida.ics@gencat.cat (L.A.-S.); sortiz.lleida.ics@gencat.cat (S.O.-C.); 2Research Group in Therapies in Primary Care (GRETAPS), Fundació Institut Universitari per a la Recerca en Atenció Primària de Salut Jordi Gol i Gurina (IDIAP JGol), 25007 Lleida, Spain; eartigues.lleida.ics@gencat.cat (E.A.-B.); mortega.lleida.ics@gencat.cat (M.O.-B.); 3Biomedical Research Institute (IRB Lleida), University of Lleida (UdL), 25198 Lleida, Spain; 4Catalan Health Institute (ICS), Primary Care Lleida, Rambla Ferran, 44, 25007 Lleida, Spain; jsol.lleida.ics@gencat.cat; 5Research Support Unit (USR), Fundació Institut Universitari per a la Recerca en Atenció Primària de Salut Jordi Gol i Gurina (IDIAP JGol), 25007 Lleida, Spain; 6Metabolic Phisiopathology Group, Department of Experimental Medicine, Biomedical Research Institute (IRB Lleida), University of Lleida (UdL), 25198 Lleida, Spain

**Keywords:** glucose metabolism disorders, circadian rhythm, sleep, blood glucose, glycated haemoglobin A1c

## Abstract

**Simple Summary:**

Since several studies have described a relationship between sleep disturbances and abnormal glucose metabolism, improving sleeping habits in people with type 2 diabetes should improve glucose metabolism. To prove this hypothesis, we conducted an educational intervention to ameliorate sleep hygiene through nine simple recommendations in patients with prediabetes and type 2 diabetes. We then evaluated if sleep quality, levels of blood glucose and glycated haemoglobin had improved. In the intervention group, we found a significant improvement in sleep quality and diabetes control compared with the control group. Education in sleep hygiene is an important tool for improving health in people with prediabetes and diabetes.

**Abstract:**

Background: Evidence supports a causal relationship between circadian disturbance and impaired glucose homeostasis. Methods: To determine the effect of an educational intervention delivered by primary care nurses to improve sleep hygiene, a parallel, open-label clinical trial in subjects aged 18 and older with impaired fasting glucose (IFG) or type 2 diabetes mellitus (T2DM) was performed. Study variables were sex, age, fasting glucose, glycated haemoglobin A1c (HbA1c), Pittsburgh Sleep Quality Index (PSQI), sleep duration and efficiency, body mass index, antidiabetic treatment, diet and physical exercise. An individual informative educational intervention was carried out following a bidirectional feedback method. The intervention aimed to develop skills to improve sleep through nine simple tips. An analysis of covariance was performed on all the mean centred outcome variables controlling for the respective baseline scores. Results: In the intervention group, PSQI dropped, the duration and quality of sleep increased, and a decrease in fasting glucose and in HbA1c levels was observed. Conclusion: The proposed intervention is effective for improving sleep quality, length and efficiency, and for decreasing fasting glucose and HbA1c levels in only 3 months. These findings support the importance of sleep and circadian rhythm education focused on improving IFG and T2DM.

## 1. Introduction

Diabetes mellitus is a chronic condition that occurs when blood glucose levels increase because the body cannot produce enough insulin or cannot effectively use the insulin it produces [1]. It is a condition of high public health concern, causing important morbidity, mortality and significant loss of quality of life [2]. It is estimated that 536.6 million adults aged 20–79 worldwide (9.8% of all adults in this age range) have diabetes. Based on 2021 estimates, 783.7 million adults aged 20–79 are expected to have diabetes in 2045. In Europe, the estimated prevalence in 2021 was about 61,425 million people, and by 2045 this number is projected to increase to 1.7% of the population, affecting 69 million people [1].

Type 2 diabetes mellitus (T2DM) is the most common type of diabetes, accounting for around 90% of all diabetes globally [3,4]. T2DM is a chronic metabolic disease that can be controlled when its pathophysiological factors are neutralised [5]. However, diabetes management remains a challenge [6]. T2DM onset is preceded by alterations in blood glucose levels known as prediabetes, which refers to the hyperglycaemic conditions of impaired fasting glucose (IFG) and impaired glucose tolerance (IGT) [7]. Hyperglycaemia is caused by the inability of the body’s cells to fully respond to insulin, a phenomenon known as insulin resistance, which causes an overproduction of insulin as a compensatory mechanism, resulting in the failure of pancreatic beta cells and a dysfunction in insulin production [8].

T2DM is associated with several risk factors, some of them unmodifiable. However, other risk factors such as overweight and obesity, sedentary lifestyle, smoking and dietary patterns can be modified [9,10]. A relevant modifiable risk factor is sleep [11,12].

Low quality of sleep is associated with an increase in cortisol, growth hormone and ghrelin levels, and a decrease in leptin levels [13]. High levels of cortisol and growth hormone have been found to interact with insulin receptors as insulin antagonists [14], while low leptin levels and high ghrelin levels are associated with an increased risk of obesity, either by reducing satiety or stimulating appetite [15]. All these factors result in an increase in insulin resistance and therefore a higher risk of developing T2DM [14,15].

Quality of sleep comprises a wide range of dimensions, including efficiency, time, sleep quality and alertness or sleepiness. These can be measured objectively through polysomnography, or subjectively through self-reports such as the Pittsburgh Sleep Quality Index (PSQI) [16,17,18]. The PSQI is considered an effective tool to assess sleep quality in T2DM [19].

Quality of sleep is considered more relevant than sleep duration. While several studies concluded that both short and long duration of sleep could play a causal role in T2DM [20,21,22], others indicate that good sleep quality could protect against this disease [23].

We already know that sleep has a restorative function that benefits glucose metabolism. Given that circadian rhythm disturbance is an environmental risk factor for T2DM [24], we should now investigate if we should emphasise quality or quantity of sleep in people with abnormal glucose tolerance.

Circadian disturbances are defined as a mismatch between the endogenous circadian system and behavioural circadian cycles (e.g., sleep–wake and fast–eat). In today’s 24 h society, circadian disturbance is becoming increasingly common, driven primarily by increased exposure to artificial lighting, rotating and nightshift work, and social jet lag [24,25].

Several lines of evidence support a causal relationship between circadian disturbance and impaired glucose homeostasis. Firstly, shift workers and night workers have a higher prevalence of diabetes, glucose intolerance and metabolic syndrome. Moreover, clinical studies conducted in controlled experimental settings show that acute exposure to circadian disturbance results in dysregulation of glucose metabolism characterised by impaired insulin secretion and action [26,27].

Several studies have been performed in order to test the efficacy of interventions to improve the quality of sleep in comorbid individuals [28,29]. However, the hypothetical effect of a complementary therapy for T2DM and prediabetes based on sleep hygiene in order to improve sleep quality has not been broadly studied [30,31]. Furthermore, all experimental studies on sleep and T2DM have focused on restricting sleep, but no studies have attempted to improve sleep quality [20]. This improvement could lead to better endocrine regulation and result in better management of the disease. The aim of the current study was to analyse the effect of a sleep hygiene intervention in the management of IFG and T2DM.

## 2. Materials and Methods

Design and study population: Experimental study based on a parallel clinical trial using blocked randomization with equal allocation ratio. The intervention was conducted by Primary Care Nursing of the Primary Health Centre of Balaguer in Catalonia (Spain). Due to the nature of the intervention, the trial was open-labelled. The study population consisted of adults (aged 18 and older) diagnosed with IFG or T2DM who attended regular nursing visits. Adults with HbA1c higher than or equal to 5.7% at the time of diagnosis (based on diagnostic criteria from the Red GDPS guideline) [32] and PSQI greater than 5 points (poor sleep quality) [17] were included. Subjects diagnosed with obstructive sleep apnoea syndrome, narcolepsy, fibromyalgia, dementia, schizophrenia, psychosis, major depression and shift workers, as well as those who refused to participate, were excluded from the study. Participants who required a change in antidiabetic treatment during the study period were not included in the analysis since they might have had significant effects on the outcome variables.

Participants were recruited through consecutive non-probability sampling. Participation was offered to all patients who attended the nursing follow-up visits until the predefined sample size was obtained.

Intervention: Individual education was carried out following a bidirectional feedback method. The education aimed to develop skills for making conscious and autonomous decisions. The education consisted of: (1) Information and reading of the educational sheet with subsequent discussion: 9 tips for a healthy sleep, according to the latest guidelines developed by the American Academy of Sleep Medicine [33], the National Health Service [34] and the Health Department of Catalonia [35]. These 9 tips emphasised that maintaining a regular and sufficient sleep schedule and establishing a series of routines and habits in the hours prior to going to sleep would prevent early awakenings. Each tip was read and discussed with the participant. (2) Verification: even if the participant did not ask any questions, the nurse asked if they had understood the advice. (3) Participant information: Questions such as “Did you already know any of these tips?” were asked to the participant. One telephone call per month was made as educational reinforcement of the intervention.

In the first visit, nursing professionals recruited the patients for the study if they were eligible. During the same visit, a PSQI test was interviewer-administered. Subjects with a PSQI result greater than 5 points (poor sleep quality) were randomly assigned either to the control or intervention group. Next, health professionals checked for the most recent laboratory records of fasting glucose and HbA1c levels of the participants, and the values were accepted if recorded within 6 months prior to the visit. Otherwise, new blood tests were performed. Anthropometric measurements, as well as information regarding diet, physical exercise and current antidiabetic pharmacological treatment, were obtained. At another visit, the individual intervention was carried out only with the participants assigned to the intervention group, and follow-up visits were scheduled three months after the first visit for both intervention and control groups. One month after the intervention, the nurse conducted educational reinforcement of sleep hygiene by phone.

On the follow-up visit (3 months after the beginning of the intervention), another PSQI test was interviewer-administered to the participants, and new blood tests were performed to assess fasting glycaemia and HbA1c. Updates about sleep hygiene, diet, physical exercise and current antidiabetic pharmacological treatment were requested. The values obtained from the blood tests corresponding to the next follow-up visit, approximately 6 months after the beginning of the intervention, were recorded. A total of three on-site visits in both groups and one telephone visit were made in the intervention group. The study did not promote any special diet nor any other treatments or therapies apart from the sleep intervention.

Variables: The main outcome variable was levels of HbA1c (%) 3 and 6 months post intervention. Secondary outcomes were fasting glucose (mg/dL) 3 and 6 months post intervention, PSQI, declared sleep hours (hours) and sleeping efficiency (number of hours the patient declared having slept divided by the number of hours the patient declared having stayed in bed multiplied by 100 and expressed in percentage) 3 months post intervention.

The independent variables were: pre-intervention values of HbA1c (%), fasting glycaemia (mg/dL), PSQI score, declared sleep hours (hours) and sleeping efficiency (%), as well as age (years), sex (man, woman), diagnosis (International Classification of Diseases, 10th version: E11 Diabetes Mellitus type 2; R73 Elevated blood glucose level), antidiabetic pharmacological treatment (yes, no), body mass index (kg/m^2^), change in antidiabetic pharmacological treatment in final visit (yes, no), change in diet in final visit (yes, no), change in physical exercise in final visit (yes, no) and change in sleep hygiene in final visit (yes, no).

Statistical analysis: Descriptive statistics were used to summarise the variables in both groups (intervention and control), data normality was evaluated using the Shapiro–Wilk test, and the results were described using medians and IQR or means and SD depending on whether the distribution was non-normal or normal, respectively. Count data were described as absolute and relative frequencies. Differences between groups were assessed with a Mann–Whitney test or a Student’s *t*-test for numeric variables depending on the data distribution, and with a Chi-squared test for count data. Changes in the outcome variables during the study period were evaluated with either a paired Mann–Whitney test or with a paired Student’s *t*-Test.

The efficacy of the intervention was assessed with an analysis of covariance on all the outcome variables controlling for the respective baseline scores. For this purpose, and since linear regression models did not fit the assumptions, quantile regressions for each variable were generated, considering the post intervention scores as the response, the group as the predictor variable and the respective baseline score as a covariate. Interactions between baseline scores and the group were included and the models were adjusted for baseline PSQI scores, age, sex, diagnosis, body mass index, changes in diet, changes in physical exercise, benzodiazepine intake and time from blood test to intervention. The regression coefficients, the corresponding 95% CI and the statistical significance were estimated.

The analysis was carried out in R version 3.4.4 (R Core Team 2018; R Core Team; R: language and environment for statistical computing; R Foundation for Statistical Computing, Vienna, Austria, 2018; URL https://www.R-project.org/ (accessed on 14 January 2020)).

## 3. Results

Recruitment and follow-up took place between November 2017 and November 2018. A total of 133 subjects were assessed for eligibility, and 69 were included in the analysis (31 and 38 from the control and intervention groups, respectively). The flow diagram of the process is specified in Figure 1, and the demographic and clinical descriptions of individuals in both groups are presented in Table 1.

Treatment changes associated with worsening T2DM were observed in both the intervention and control groups, but the number of treatment changes in the control group was significantly higher (28% control group versus 7% intervention group, *p* < 0.005). These changes could impact fasting glucose and HbA1c results. For this reason, these subjects were not analysed.

Clinical and demographic variables, habit changes and baseline outcome variables were studied to assess group comparability (Table 1 and Table 2). Small imbalances were observed between groups regarding all the variables except for benzodiazepine intake.

**Effect of the intervention on sleep parameters:** 84.2% participants from the intervention group and 14.0% in the control group reported a change in sleep habits. 3 months after the intervention, the control group did not report any change in the PSQI (−0.61 ± 3.11), hours of sleep (0.00 [−0.50; 1.00] hours) or sleep efficiency (−1.65 ± 12.0%), while the intervention group reported a statistically significant improvement in all three parameters: PSQI (−2.97 ± 2.93), hours of sleep (1.00 [0.00; 2.00] hours) and sleep efficiency (6.74 ± 12.9%) (Table 2). A significantly higher number of subjects in the intervention group reported more than 6 h of sleep and an improvement of 3 or more points in the PSQI (Table 3).

Moreover, when comparing both groups, the intervention group obtained lower post-intervention PSQI scores (−3.62; 95% CI: −5.73, −1.51), longer perceived sleep time (1.54; 95% CI: 0.79, 2.28) and higher sleep efficiency (9.78; 95% CI: 1.95, 17.61) (Figure 2). The effect of the intervention was the same across all baseline scores (interaction *p*-value > 0.05).

**Effect of the intervention on T2DM management:** When assessing the effect on T2DM management, in the control group no variation was observed either in fasting glucose (3.00 [−10.50; 13.0] mg/dL) nor HbA1c (0.20 [−0.20; 0.55] %) levels 3 months after the intervention. However, the intervention group experienced a decrease in both fasting glucose (−12.50 [−27.00; 1.00] mg/dL) and HbA1c (−0.20 [−0.50; −0.02] %) (Table 2).

6 months post intervention, no changes were observed in the control group regarding fasting glucose (4.43 ± 22.1 mg/dL), and higher HbA1c values were observed (0.42 ± 0.58%). In the intervention group, no changes were detected in fasting glucose (−4.08 ± 25.8 mg/dL) or HbA1c (−0.06 ± 0.64%) (Table 2). Furthermore, 6 months after the intervention, a higher proportion of subjects with an improvement of more than 0.5% in HbA1c levels was observed (Table 3).

When comparing both groups using quantile regression models, the intervention group achieved a significant reduction in 3 months post intervention fasting glucose levels (−14.69; CI 95%: −28.15, −1.22) and HbA1c levels (−0.39; 95% CI: −0.73, −0.05) in relation to the control group, as well as a reduction in 6 months post intervention HbA1c levels (−0.66; 95% CI: −0.96, −0.36) (Figure 3). The effect of the intervention was the same across all baseline levels (interaction *p*-value > 0.05).

**Relationship between change in PSQI and T2DM management:** Finally, the effect of the changes in PSQI during the intervention period was tested. No relationship between PSQI changes and fasting glucose changes was found. However, we found a statistically significant association between the change in HbA1c levels after 3 months and the change in PSQI (0.06; 95% CI: 0.02, 0.11) (Figure 4).

## 4. Discussion

Several studies have found a causal relationship between the deterioration in the quality and duration of sleep and T2DM onset [36], as well as impaired glucose metabolism under sleep restriction [37,38]. However, to our knowledge, there are no studies on interventions aiming to improve the sleep quality of patients with T2DM to regulate glucose homeostasis. 

The objective of the current study was to evaluate the effectiveness of a sleep hygiene intervention on sleep quality, and to assess whether this improvement resulted in a decrease in glucose and HbA1c levels.

The results show a significant improvement in all the measured sleep parameters (sleep quality, time and efficiency). Moreover, the PSQI score improvement in the intervention group was 3.6 points higher compared to the control group. An improvement of ≥3 points in the PSQI has been defined as a positive response to treatment [39], supporting the clinical relevance of the results. The median sleep time in the intervention group was increased by 1.5 h, doubling the number of participants who reported sleeping more than 6 h. Thanks to this improvement, more than 50% of the participants in the intervention group slept 6 h or more after the intervention, which is the minimum recommended sleep time in adults [40,41]. Shorter sleep times have been related to a higher risk of developing T2DM [42].

Beyond the behavioural effect that hygiene measures have on improving the quality and duration of sleep, the intervention could have enhanced circadian contrast, i.e., being exposed to light during the day and to the dark at night and fasting at night and eating during the day. The regularity of sleep and sleep/wake patterns and the magnitude of the difference between daytime and night-time activity measured with technology based on temperature register has shown a decrease in HbA1c [43].

Regarding T2DM management, a significant improvement was also observed in fasting glucose and HbA1c levels. HbA1c changes in the intervention group (reductions of 0.39% and 0.66% at 3 and 6 months, respectively) are comparable to those obtained through nutritional interventions [44] and second line treatments [45]. The main endpoint to establish clinical relevance is a reduction of at least 0.5% [46], and a 0.3% endpoint has also been used for evaluating both behavioural interventions and second line treatment effectiveness [47], indicating the short- and mid-term success of the intervention. The effectiveness of the intervention is further reinforced by the significant correlations between pre–post intervention changes in PSQI with pre–post intervention changes in HbA1c. Interestingly, the generated model intercepts with the 0.0 point, indicating that positive changes in PSQI correlate with positive changes in HbA1c, and vice versa.

These results confirm the importance of sleep in relation to the circadian rhythm of glucose metabolism [47]. This health education has improved the preparation and maintenance of sleep in the intervention group, resulting in longer and better sleep, until it has become a consolidated zeitgeber. Healthy sleep patterns allow the 24 h oscillation in glucose levels attributable to the nightly rest and fast to influence the expression of circadian genes and rhythmic transcriptional outputs within the hypothalamic neurons involved in glucose homeostasis [23].

Notably, there was a slight (but not statistically significant) improvement in sleep quality in the control group. This might be in response to the information offered to the participants as part of the informed consent, which stressed the importance of sleep in people with T2DM or IFG. Nevertheless, participants in the intervention group, with access to specific guidelines, increased sleep quality to a greater extent.

Primary care nurses have a decisive role in monitoring and managing chronic diseases such as T2DM. For this reason and in order to ensure its applicability, the intervention was integrated into the regular patient check-ups. The current study demonstrates that a simple sleep health educational intervention delivered during the regular check-ups of patients with T2DM or IFG has a positive metabolic effect and is feasible as a complementary therapy in primary care settings. Our study has some limitations. Firstly, the intervention period was short, and adherence to educational interventions is known to decrease over time [48,49,50]. However, we have demonstrated that following the guidelines has short-term rewarding effects, and nurses can regularly provide diet and physical exercise recommendations that could increase adherence and positive reinforcement. Secondly, since we obtained positive results in only 3 months, we would have preferred to use baseline parameters closer to the start of the intervention, which was not possible due to laboratory limitations. However, we found similar distributions between groups regarding time from blood test to the initial visit, and significant correlations between PSQI changes and HbA1c changes, supporting the reliability of the results. Finally, taking into account that the intervention was open-labelled, the use of subjective variables related to sleep quality (instead of objective tests such as polysomnography) could lead to biased self-reports in the intervention group. However, the PSQI is considered a valid tool that incorporates all dimensions of sleep and is widely used for assessing sleep quality in patients with T2DM [51,52].

## 5. Conclusions

An educational intervention in sleep hygiene and circadian contrast delivered by primary care nurses increased sleep quality as measured with the PSQI. This intervention lowered HbA1c levels in patients with IFG and T2DM. Sleep education improves T2DM metabolic management.

## Figures and Tables

**Figure 1 biology-11-00893-f001:**
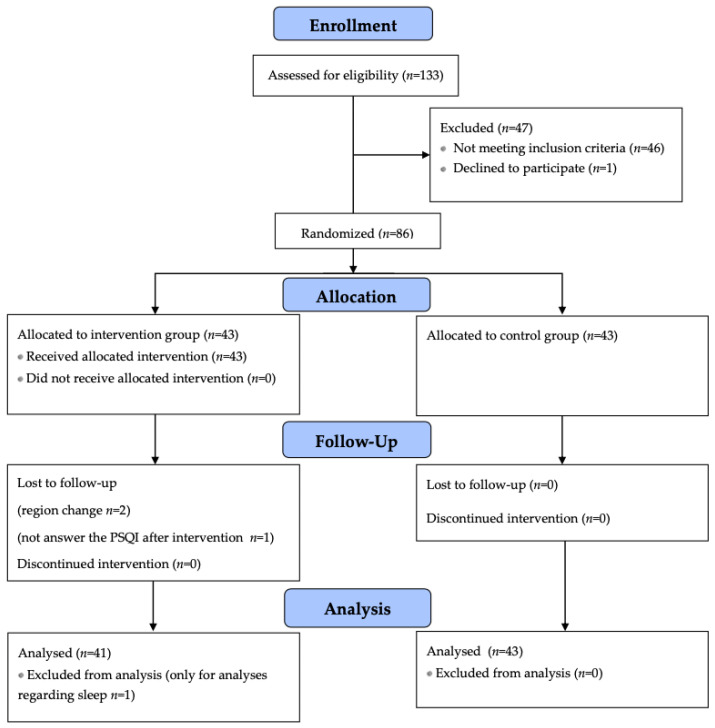
Flow diagram of the selection and follow-up process.

**Figure 2 biology-11-00893-f002:**
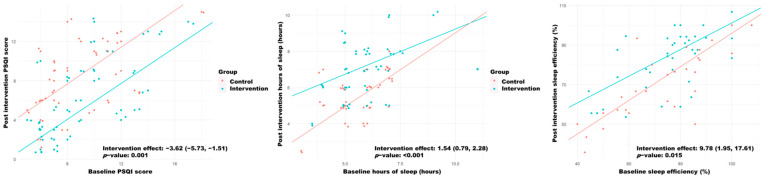
Quantile regression models for the response variables “Post-intervention hours of sleep” and “Post-intervention sleep efficiency” as a function of the group and the respective baseline scores. Each point represents the participant’s observed score, and the lines represent the fitted models. The intervention effects are shown, with their corresponding 95% confidence interval and *p*-value.

**Figure 3 biology-11-00893-f003:**
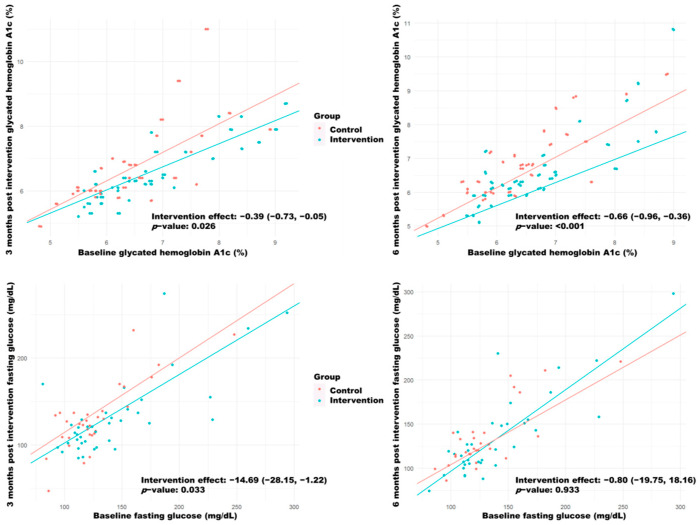
Quantile regression models for the response variables “Post-intervention HbA1c” and “Post-intervention fasting glycemia” evaluated at 3 and 6 months as a function of the group and the respective baseline scores. Each point represents the participant’s observed score, and the lines represent the fitted models. The intervention effects are shown, with their corresponding 95% confidence interval and *p*-value.

**Figure 4 biology-11-00893-f004:**
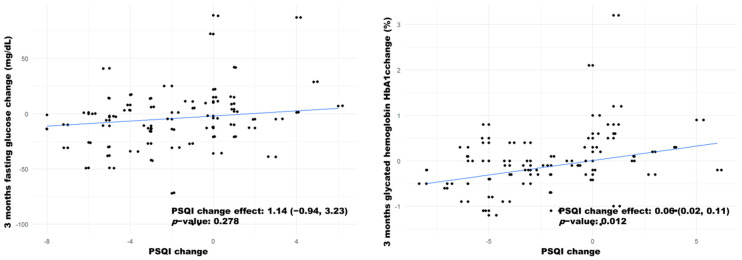
Quantile regression models for the response variables “3 months glucose change” and “3 months HbA1c change” as a function of PSQI changes. Each point represents the participant’s observed score, and the lines represent the fitted models. The PSQI change effects are shown, with their corresponding 95% confidence interval and *p*-value.

**Table 1 biology-11-00893-t001:** Demographic and clinical characteristics of the intervention and control group.

Variable	Control Group (*n* = 31)	Intervention Group (*n* = 38)	Total (*n* = 69)
Age (years)	64.5 (12.9)	66.5 (11.3)	65.6 (12.0)
Body Mass Index (Kg/m^2^)	31.0 (5.6)	29.9 (4.1)	30.4 (4.8)
Time from blood test to intervention (days)	46.0 [19.5;108]	101 [15.0;138]	62.0 [18.0;133]
Sex (woman)	15 (48.4%)	22 (57.9%)	37 (53.6%)
Diagnosis (T2DM)	29 (93.5%)	33 (86.8%)	62 (89.9%)
Antidiabetic pharmacological treatment (yes)	25 (80.6%)	31 (81.6%)	56 (81.2)
Benzodiazepine intake (yes)	5 (16.1%)	10 (26.3%)	15 (21.7%)
Preintervention sleep time > 6 h (yes)	9 (29.0%)	11 (28.9%)	20 (29.0%)
Diet change (yes)	2 (6.45%)	4 (10.5%)	6 (8.70%)
Physical activity change (yes)	3 (9.68%)	1 (2.63%)	4 (5.80%)

T2DM, type 2 diabetes mellitus.

**Table 2 biology-11-00893-t002:** Comparison of quantitative variables in the control group versus the intervention group before and after the intervention.

Response Variables	Pre Values			Post Values			Change	
	Control Group	Intervention Group	*p* Value	Control Group	Intervention Group	*p* Value	Control Group	Intervention Group
	Sleep							
PSQI	8.00 [6.00;11.0]	8.00 [6.00;10.8]	0.98	8.00 [6.00;11.0]	5.00 [3.00;9.00]	**0.008**	−0.61 (3.11)	−2.97 (2.93) **
Hours of sleep	6.00 [5.00;6.75]	6.00 [5.00;6.50]	0.76	6.00 [5.00;6.50]	7.00 [6.00;8.00]	**0.002**	0.00 [−0.50;1.00]	1.00 [0.00;2.00] *
Sleep efficiency	75.0 [58.0;86.3]	77.3 [66.7;84.0]	0.809	66.7 [57.1;84.5]	85.7 [71.4;93.3]	**0.007**	−1.65 (12.0)	6.74 (12.9) **
	T2DM management							
Fasting glucose (3 months)	122 [106;134]	126 [112;154]	0.249	127 [112;138]	121 [102;137]	0.473	3.00 [−10.50;13.0]	−12.50 [−27.00;1.00] *
Fasting glucose (6 months)	122 [106;134]	126 [112;154]	0.249	122 [114;140]	120 [106;151]	0.766	4.43 (22.1)	−4.08 (25.8)
HbA1c (3 months)	6.40 [5.85;7.10]	6.45 [5.90;7.15]	0.443	6.40 [6.00;7.10]	6.25 [5.82;7.15]	0.476	0.20 [−0.20;0.55]	−0.20 [−0.50;−0.02] *
HbA1c (6 months)	6.40 [5.85;7.10]	6.45 [5.90;7.15]	0.443	6.75 [6.00;7.43]	6.30 [5.95;7.02]	0.315	0.42 (0.58) **	−0.06 (0.64)

* Statistically significant (paired Mann–Whitney test, *p* < 0.05); ** Statistically significant (paired Student’s *t* test, *p* < 0.05); PSQI, Pittsburgh Sleep Quality Index; T2DM, type 2 diabetes mellitus; HbA1c, glycated haemoglobin A1c.

**Table 3 biology-11-00893-t003:** Comparison of qualitative variables in the control group in relation to the intervention group before and after the intervention.

Response Variables	Control Group (*n* = 31)	Intervention Group (*n* = 38)	Total	*p* Value
Sleep hygiene change (yes)	5 (16.1%)	32 (84.2%)	37 (53.6%)	**<0.001**
PSQI change ≤ −3 (yes)	9 (29.0%)	22 (59.5%)	31 (45.6%)	**0.024**
Post intervention sleep time > 6 h (yes)	9 (29.0%)	21 (56.8%)	30 (44.1%)	**0.041**
3 months HbA1c change ≤ −0.5% (yes)	4 (12.9%)	13 (34.2%)	17 (24.6%)	0.078
6 months HbA1c change ≤ −0.5% (yes)	1 (3.57%)	10 (29.4%)	11 (17.7%)	**0.009**

PSQI, Pittsburgh Sleep Quality Index; HbA1c, glycated haemoglobin A1c.

## Data Availability

The data are hosted on the research team’s internal servers and will be provided under reasonable request.

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
