# Peer review of "Enhancing Night and Day Circadian Contrast through Sleep Education in Prediabetes and Type 2 Diabetes Mellitus: A Randomized Controlled Trial"

_biology, 2022, doi:10.3390/biology11060893_

Round 1
Reviewer 1 Report
The research by García-Serrano et al. is exciting and well-designed, although it was open-labeled. Although fewer subjects were included in the study than anticipated, the authors obtained interesting results and showed how little investment in patient education could contribute to improving their health.
Some minor comments are:
Line 72-73 - the sentence should be written more clearly.
As the authors mentioned in lines 160–162, an estimated sample size of 84 participants was required to obtain a study strength of 0.8. Therefore, the authors should calculate the actual strength of the study.
Some grammar and spelling errors should be corrected (e.g., lines 41, 100, etc.).
Author Response
Comment (C):
The research by García-Serrano et al. is exciting and well-designed, although it was open-labeled. Although fewer subjects were included in the study than anticipated, the authors obtained interesting results and showed how little investment in patient education could contribute to improving their health.
Some minor comments are:
Line 72-73 - the sentence should be written more clearly.
Answer (A):
We have rewritten the sentence, as follows:
“We already know that sleep has a restorative function that benefits glucose metabolism. In view that circadian rhythm disturbance is an environmental risk factor for T2DM, [24] we should now investigate if we should emphasize quality or quantity of sleep in people with abnormal glucose tolerance.”
C:
As the authors mentioned in lines 160–162, an estimated sample size of 84 participants was required to obtain a study strength of 0.8. Therefore, the authors should calculate the actual strength of the study.
A:
We have performed a simulation-based post-hoc power analysis for our main outcome (post-intervention HbA1c levels) using the actual distribution of the variables and we have obtained a power of 0.74 for detecting a reduction of 0.5 on HbA1c levels with a 95% of confidence using the multivariate quantile regression. However, we have not included this information in the manuscript since post-hoc power analyses are not recommended (Goodman S N, and Berlin J A.Ann Intern Med. 1994 Aug 1;121(3):200-6). In order to avoid misunderstandings, we have also removed the sample size calculation from the main manuscript.
C:
Some grammar and spelling errors should be corrected (e.g., lines 41, 100, etc.).
A:
The english redaction has now been reviewed by a professional translator. We hope that this has improved the quality of the english redaction of the new version of the manuscript.
Reviewer 2 Report
Dear authors,
I carefully read your manuscript, and I have some doubts or suggestions, which, I hope, will improve the quality of your article.
My major concerns are about the sample size, which is insufficient to fulfil the statistic power and the fact that it is unclear if subjects underwent a special diet or started a drug therapy. If so, diet, pharmacology therapy, and also physical activity could have influenced HbA1c results.
You state that you investigated physical activity; however, it is unclear if you used this variable as a covariate or considered it in the analyses.
Furthermore, the article seems not to completely fulfil and follow the journal's aim since the authors neither investigate biological mechanisms nor explain potential linkages between sleep and diabetes mellitus. In my opinion, the authors should reinforce this aspect in the discussion section; otherwise, this article would be useful in this journal.
Please, see later for minor comments:
1. Line 43: abbreviation should be inserted the first time you name the entire word. Is type 2 diabetes mellitus different from diabetes mellitus? In my opinion, it is better firstly to describe the pathology and then explain how it will grow in the future years.
2. Lines 62 and 92 are missing the references.
3. In line 79, I suggest inserting the following articles: Galasso et al., 2021, doi: 10.3390/ijerph18168378.
4. What do you mean with hetero-administrated?
5. Line 152: maybe some independent variables listed here could be considered covariates.
6. You should explain if participants start special diets, therapies or other treatments.
7. Sometimes you use the abbreviation T2DM and sometimes T2D. Why?
8. In table 2, DM2 is not explained.
9. Lines 234-237: are the results shown in figure 3? Are these results obtained with the quantile regression model?
10. The figures are of low quality.
Author Response
Comment (C):
Dear authors,
I carefully read your manuscript, and I have some doubts or suggestions, which, I hope, will improve the quality of your article.
My major concerns are about the sample size, which is insufficient to fulfil the statistic power
A:
We have performed a simulation-based post-hoc power analysis for our main outcome (post-intervention HbA1c levels) using the actual distribution of the variables and we have obtained a power of 0.74 for detecting a reduction of 0.5 on HbA1c levels with a 95% of confidence using the multivariate quantile regression. However, we have not included this information in the manuscript since post-hoc power analyses are not recommended (Goodman S N, and Berlin J A.Ann Intern Med. 1994 Aug 1;121(3):200-6). In order to avoid misunderstandings, we have also removed the sample size calculation from the main manuscript.
C:
and the fact that it is unclear if subjects underwent a special diet or started a drug therapy. If so, diet, pharmacology therapy, and also physical activity could have influenced HbA1c results.
A:
The study did not promote any special diet, other treatments or therapies apart from the sleep hygiene intervention. Those patients that changed their antidiabetic treatment were excluded from the analysis, as stated in Figure 1. On the other hand, the information about diet and physical activity changes is reported in Table 1. We did not collect any information about changes in other non-pharmacological therapies.
We have added this information in the methods section of new version of the manuscript:
“A total of three on-site visits in both groups, and one telephone visit were made in the intervention group. The study did not promote any special diet, other treatments or therapies apart from the sleep intervention.”
In order to account for the effect of the potential confounding variables, we have redone the analyses using basal PSQI scores, age, sex, diagnosis, body mass index, changes in diet, changes in physical exercise, benzodiazepine intake and time form blood test to intervention as covariates. We have also taken into account the interaction between the group and the basal scores.The results are almost the same and all significances are maintained. We have modified the figures and the manuscript with the new adjusted values using these covariates. We have also specified it in the methodology section, as follows:
“The efficacy of the intervention was assessed with an analysis of covariance on all the outcome variables controlling for the respective baseline scores. For this purpose, and since linear regression models did not fit the assumptions, quantile regressions for each variable were generated, considering the post intervention scores as the response, the group as the predictor variable and the respective baseline score as a covariate. Interactions between baseline scores and the group were included and the models were adjusted for baseline PSQI scores, age, sex, diagnosis, body mass index, changes in diet, changes in physical exercise, benzodiazepine intake and time from blood test to intervention. The regression coefficients, the corresponding 95% CI and the statistical significance were estimated.”
C:
You state that you investigated physical activity; however, it is unclear if you used this variable as a covariate or considered it in the analyses.
A:
We have now used it as a covariate. We have specified it in the methodology section.
C:
Furthermore, the article seems not to completely fulfil and follow the journal's aim since the authors neither investigate biological mechanisms nor explain potential linkages between sleep and diabetes mellitus. In my opinion, the authors should reinforce this aspect in the discussion section; otherwise, this article would be useful in this journal.
A:
We have reinforced these aspects in the discussion section of the new version of the manuscript, as follows:
“These results confirm the importance of sleep in relation to the circadian rhythm of glucose metabolism [45] [46]. This health education has improved the preparation and maintenance of sleep in the intervention group, resulting in longer and better sleep, until it has become a consolidated zeitgeber. Healthy sleep patterns allow the 24-hour oscillation in glucose levels attributable to the nightly rest and fast influence the expression of circadian genes and rhythmic transcriptional outputs within the hypothalamic neurons involved in glucose homeostasis [45]. ”
C:
Please, see later for minor comments:
- Line 43: abbreviation should be inserted the first time you name the entire word. Is type 2 diabetes mellitus different from diabetes mellitus? In my opinion, it is better firstly to describe the pathology and then explain how it will grow in the future years.
A:
We have corrected it in the new version of the manuscript, as follows:
“Diabetes mellitus is a chronic condition that occurs when blood glucose levels in-crease because the body cannot produce enough insulin, or cannot effectively use the in-sulin it produces [2]. It is a condition of high public health concern, causing important morbidity, mortality and significant loss of quality of life [1]. It is estimated that 536.6 million adults aged 20 to 79 years worldwide (9.8% of all adults in this age range) have diabetes. Based on 2021 estimates, 783.7 million adults aged 20 to 79 years are expected to have diabetes in 2045. In Europe, the estimated prevalence in 2021 was about 61,425 million people, and by 2045 this number is projected to increase to 1.7%, affecting 69 mil-lion people [2].“
C:
- Lines 62 and 92 are missing the references.
A:
We have corrected it in the new version of the manuscript
C:
- In line 79, I suggest inserting the following articles: Galasso et al., 2021, doi: 10.3390/ijerph18168378.
A:
We thank the reviewer for the suggestion. We have included the reference in the new version of the manuscript.
C:
- What do you mean with hetero-administrated?
A:
We apologize for the misuse of the word. We referred to a questionnaire in which the professional is the one who asks the questions from the test to the patient and collects the information. We have changed it for interviewer-administered.
C:
- Line 152: maybe some independent variables listed here could be considered covariates.
A:
As we have stated in a previous comment, we have redone the analyses using basal PSQI scores, age, sex, diagnosis, body mass index, changes in diet, changes in physical exercise, benzodiazepine intake and time form blood test to intervention as covariates and modified the manuscript accordingly.
C:
- You should explain if participants start special diets, therapies or other treatments
A:
As we have stated in a previous comment, the study did not promote any special diet, other treatments or therapies apart from the sleep hygiene intervention. Those patients that changed their antidiabetic treatment were excluded from the analysis, as stated in Figure 1. On the other hand, the information about diet and physical activity changes is reported in Table 1. We did not collect any information about changes in other non-pharmacological therapies.
C:
- Sometimes you use the abbreviation T2DM and sometimes T2D. Why?
A:
We apologize for the mistake. We have amended it in the new version of the manuscript.
C:
- In table 2, DM2 is not explained.
A:
We have changed it for T2DM and we have described the abbreviation in the new version of the manuscript. We have also described the other abbreviations of the table.
C:
- Lines 234-237: are the results shown in figure 3? Are these results obtained with the quantile regression model?
A:
These results are indeed obtained with the quantile regression model. We have changed into “When comparing both groups using quantile regression models,...“
C:
- The figures are of low quality.
A:
We have modified the figures in order to improve their quality and according to the new analyses. We have also added a small artificial variability to those graphs with overlapped points in order to improve visualization.